# STREAMING SPATIAL-TEMPORAL PROMPT LEARNING FOR RGB-T TRACKING

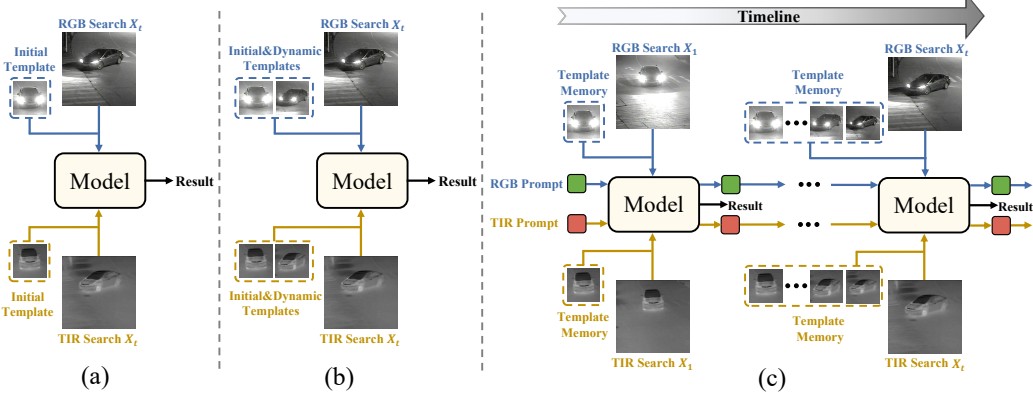

Figure 1: **Different RGB-T tracking methods**. (a) Non-temporal methods. (b) Temporally-sparse methods by introducing an additional dynamic template. (c) Our proposed streaming temporal method based on historical templates via prompt learning.

## ABSTRACT

In the process of multimodal interaction, effective spatial-temporal information of correlated targets is crucial for RGB-T tracking. However, most existing methods only utilize spatial information for template-search matching or merely introduce an additional dynamic template with sparse temporal perception. These approaches overlook rich temporal cues across consecutive video frames, such as target appearance changes and motion trajectory. To establish effective spatial-temporal associations during multimodal interaction, we propose a video-level RGB-T tracking paradigm via prompt learning, termed **PromptTrack**. It densely models the spatial-temporal relationships of targets in multimodal contexts by incorporating streaming spatial-temporal prompts within a continuous sequence of video frames. Specifically, PromptTrack learns target changes and motion trajectory from historical frames through streaming temporal prompt for each modality, and then learns multimodal spatial prompt conditioned on temporal prompt to effectively leverage multimodal complementary information. Benefiting from the proposed spatial-temporal prompt learning method, PromptTrack exhibits superior target location capability and robustness in complex tracking scenarios. The novel prompt-based tracking paradigm can also be effortlessly extended to other tracking domains such as RGB-D and RGB-E. Extensive experiments on three prevailing benchmark datasets demonstrate our method achieves new state-of-the-art performances. In particular, PromptTrack achieves Precision score of 76.2% and Success score of 60.7% on LasHeR dataset while running at a real-time speed of 35 FPS. Codes and models will be released.

## 1 INTRODUCTION

RGB-T tracking is a fundamental task in visual object tracking (VOT), which aims to continuously locate a target in subsequent frames of multimodal video stream, typically given its initial bounding box in the first frame. The superior capability of leveraging complementary information from

visible (RGB) and infrared (TIR) images to handle more complex tracking scenarios, such as low illumination, adverse weather conditions, and cluttered backgrounds, has attracted significant attention from researchers. It is widely applied in various fields, including autonomous driving (Zhang et al., 2015), video surveillance (Tian et al., 2005), and robotic vision (Itti, 2004).

The key to RGB-T tracking is how to effectively explore spatial-temporal information in a multimodal context. The complementary spatial information from aligned multimodal data helps identify the precise location of the target, while temporal information from the video stream provides insight into appearance changes and motion trajectory of the target over time. According to the extent of utilizing spatial and temporal information, existing RGB-T tracking methods can be roughly classified into *non-temporal* and *temporally-sparse* methods, as illustrated in Figure 1 (a) and (b). Non-temporal methods focus on utilizing complementary spatial information during template-search matching. TBSI (Hui et al., 2023) exploits templates as the bridge to target-relevant contexts, enabling cross-modal spatial interaction between RGB and TIR search regions. ViPT (Jiawen et al., 2023) generates spatial prompts for RGB template and search images using an extra TIR-modality prompter. This paradigm significantly advances the development of RGB-T tracking but may struggle with major target appearance changes due to relying solely on the initial template. Therefore, some methods (Figure 1 (b)) introduce an additional dynamic template to enhance tracking robustness against target appearance changes. STMT (Sun et al., 2024) and TATrack (Wang et al., 2024) attempt to fuse initial and dynamic multimodal templates to construct a unified target representation to adapt to certain appearance changes. Despite achieving improvement, this temporally-sparse paradigm often fails in situations where the target is occluded or similar distractors are present, due to its substantial reliance on appearance information.

Drawing inspiration from the way humans perceive and track targets in complex environments—by relying on continuous changes of the target across consecutive video frames—we aim to leverage historical information and complementary spatial information for RGB-T tracking. To this end, we present a novel *video-level RGB-T tracking paradigm via prompt learning*, termed **PromptTrack**, to establish effective spatial-temporal associations during multimodal interaction. As illustrated in Figure 1 (c), RGB and TIR prompts (green and red blocks) are introduced into the model to learn target changes and motion trajectory from dense historical frames that are stored in the template memory. These prompts provide prior information to guide spatial-temporal associations of targets in multimodal contexts and are propagated within a continuous sequence of video frames over time.

To implement the above paradigm, we design a multimodal tracking framework based on spatial-temporal prompt learning. Specifically, a distinct group of learnable tokens as temporal prompts are incorporated into the input for RGB and TIR modalities. The historical template images and current search images are patched into images tokens. These image tokens and prompt tokens are concatenated and input into a multimodal encoder for feature extraction and multimodal interaction. During the process of the multimodal encoder, the temporal information about targets is learned during the interaction through self-attention transformer blocks (Vaswani et al., 2017) within each modality. The complementary spatial information is effectively leveraged through multimodal spatial prompt generation blocks, which also utilize temporal prompts. The interacted temporal prompts are propagated for the next timestep with the video streaming. The intra-modal relationship modeling and inter-modal interaction facilitate the thorough exploration of multimodal spatial-temporal information. Benefiting from the proposed method, PromptTrack exhibits superior target location capability and robustness in complex multimodal tracking scenarios, effectively mitigating issues such as low illumination and distractors. Due to the generality of token forms, the novel tracking framework can also be effortlessly extended to other tracking domains such as RGB-D and RGB-E. We conduct extensive experiments to demonstrate the effectiveness and scalability of our method. The experimental results show that our method achieves significant improvements in tracking performance across various complex scenarios, such as +6.0% in Precision and +4.4% in Success on the LasHeR dataset compared to the most advanced tracker TATrack (Wang et al., 2024).

In summary, the contributions of our work are as follows: (1) A novel video-level RGB-T tracking paradigm via prompt learning is proposed to establish multimodal spatial-temporal associations, which can also be extended to RGB-D and RGB-E domains. (2) An effective multimodal tracking framework is designed by utilizing streaming temporal prompt and multimodal spatial prompt for precise target location in complex scenarios. (3) Extensive experiments demonstrate the effectiveness of our method, achieving new state-of-the-art performance on three prevailing benchmark datasets.

## 2 RELATED WORK

### 2.1 RGB-T TRACKING

In past years, RGB-T tracking methods have shifted from siamese-based architectures to transformer-based architectures. Benefiting from the one-stream transformer encoder (Ye et al., 2022) for joint feature extraction and template-search matching, researchers have tried to fuse multimodal templates and search images to exploit complementary information from RGB and TIR modality. TBSI (Hui et al., 2023) achieves cross-modal interaction of search images by using a fused templates as the bridge. UnTrack (Wu et al., 2024) learns RGB and TIR common latent space through low-rank factorization and reconstruction techniques. BAT (Cao et al., 2024) designs a bi-directional adapter on top of ViPT (Jiawen et al., 2023) to mutually enhance cross-modal interaction. However, these methods focus on utilizing spatial information for multimodal template-search matching, overlooking rich temporal cues across consecutive video frames. Recent methods introduce an additional dynamic template to strengthen the robustness against significant target appearance changes. TATrack (Wang et al., 2024) constructs a basic template-search matching branch with the initial template and an online branch with the dynamic template, enabling template interaction to embed with temporal information. STMT (Sun et al., 2024) samples a dynamic template from the previous frame and enables search regions to interact with both the initial template and the dynamic template through cross-attention mechanism. Despite certain improvements, these temporally-sparse methods still struggle in complex situations such as heavy occlusion, motion blur, and similar distractors. In contrast, PromptTrack densely learns both appearance changes and motion trajectory cross consecutive frames in multimodal contexts, serving as prior guidance for the current frame to eliminate distractors.

### 2.2 PROMPT LEARNING

Prompt learning has demonstrated significant potential in enhancing model understanding of tasks by incorporating learnable prompts in computer vision. CoOp (Zhou et al., 2022b) and CoCoOp (Zhou et al., 2022a) embed learnable context prompts into the input data to help the vision model better capture contextual information. MaPLe (Khattak et al., 2023) leverages multimodal prompts to improve the model's generalization capability in image recognition. These methods focus on learning fixed contextual prompts, whereas we aim to learn spatial-temporal target information by dynamic prompts as the video stream progresses. In RGB tracking, EVPTrack (Shi et al., 2024) employs generated tokens from the initial template to propagate information across frames. HIPTrack (Cai et al., 2024) directly generates temporal prompts based on historical search features. These approaches typically require complex generation and interaction modules of temporal tokens. In contrast, our proposed temporal prompt tokens are directly inserted into the input token sequence for each modality, obviating the need for additional modules. This streamlined design reduces our framework's complexity in multimodal environments, while achieving notable performance improvements.

For multimodal tracking task, ViPT (Jiawen et al., 2023) learns modality-related prompts to adapt frozen RGB-modality models through spatial fovea operations. OneTracker (Hong et al., 2024) regard the multimodal information as a kind of prompt and provide dominant RGB tracker with additional modality-specific information in a prompt-tuning manner. QueryNLT (Shao et al., 2024) proposes a visual-language tracking framework with joint appearance and language prompt modulation, leveraging the complementarity between historical visual cues and language expressions. Similarly, we design a multimodal spatial prompt generation module to exploit the alignment characteristics of search images, with temporal prompts guiding the generation process. Notably, our spatial prompts are bidirectional, fully leveraging complementary information from RGB and TIR modalities.

## 3 METHOD

### 3.1 PROBLEM FORMULATION

Given an initial bounding box $B_0$ of the target, the initial RGB and TIR template images $Z_0 = (Z_0^v, Z_0^i)$ [1] are cropped from the first frame of a video stream. The previous *non-temporal* methods model the tracking task as $\mathcal{T} : \{Z_0, X_t\} \rightarrow B_t$, where $\mathcal{T}$ is the learned tracker that predicts the

---

[1] Here and in the following text, the superscripts $v$ and $i$ denote RGB and TIR modalities, respectively.

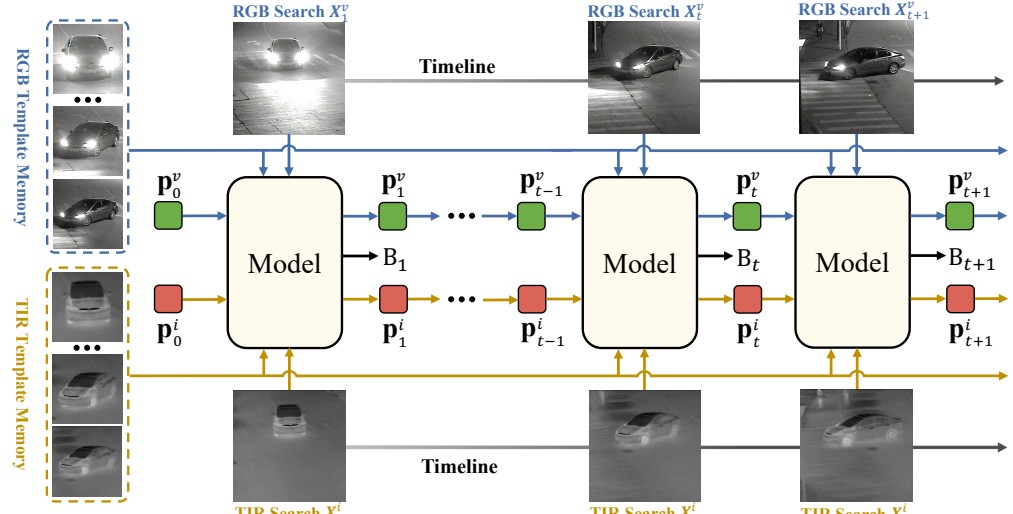

Figure 2: **The tracking pipeline across the timeline**. Streaming temporal prompts for each modality are incorporated into the input of the current timestep (search images and historical template images) and continuously updated for the next timestep. The detailed structure of **Model** is shown in Figure 3.

bounding box $B_t$ of the target in subsequent search frames $X_t = (X_t^v, X_t^i)$ at timestep $t$. In order to capture appearance changes, some *temporal-based* methods introduce dynamic template images $Z_d = (Z_m^v, Z_m^i)$, which are cropped from the middle frames of the video stream at timestep $m$ $(0 < m < t)$ and updated based on certain criteria such as confidence score and update interval. Accordingly, the temporal-based trackers can be formulated as $\mathcal{T} : \{Z_0, Z_m, X_t\} \rightarrow B_t$. However, these above trackers primarily focus on exploring multimodal spatial fusion between RGB and TIR images within sparse frames, overlooking rich temporal cues on successive video frames.

To fully mine temporal information, we redefine the multimodal tracking task as follows:

$$\mathcal{T} : \{Z_0, Z_1, ..., Z_{t-1}, X_t, P_{t-1}\} \rightarrow \{B_t, P_t\} \qquad (1)$$

This formulation models the video-level tracking process across all historical frames and incorporates streaming temporal prompts, denoted as $P = (P^v, P^i)$, to provide prior information of appearance variations and motion trends. These prompts are continuously generated and updated throughout the video stream.

### 3.2 STREAMING TEMPORAL PROMPT LEARNING

**Revisiting.** Benefiting from the powerful capabilities of the self-attention mechanism (Vaswani et al., 2017), most top-performing trackers employ the one-stream paradigm for feature extraction and relationship modeling. Specifically, for each modality, both the initial template image $Z_0$ and search image $X$ are initially segmented into non-overlapping patches, flattened, projected into template tokens $\mathbf{z}_0 = [z_1, ..., z_{N_z}] \in \mathbb{R}^{N_z \times D}$ and search tokens $\mathbf{x} = [x_1, ..., x_{N_x}] \in \mathbb{R}^{N_x \times D}$, where $N_z$ and $N_x$ denote the respective number of tokens for each image, and $D$ represents the dimension. These tokens are concatenated into a sequence of template-search tokens $[\mathbf{z}_0; \mathbf{x}] = [z_1, ..., z_{N_z}, x_1, ..., x_{N_x}] \in \mathbb{R}^{(N_z + N_x) \times D}$, and then fed into an $L$-layer modality-shared transformer encoder. Ultimately, the extracted search features from both modalities are passed to the head for prediction. This process leverages the similarity of target-relevant features to perform template-search matching solely on the spatial dimension within sparse frames, thereby highlighting a gap in video object tracking across consecutive frames.

**Evolution.** We lift the sparse-frame matching paradigm to video-level relationship modeling by utilizing historical frames and extend it with temporal prompt learning. The overall tracking pipeline is illustrated in Figure 2. As the video stream progresses, each frame is cropped into an intermediate template image $\mathbf{z}_m$ $(0 < m < t)$ according to the predicted bounding box and then stored in the historical **t**emplate **m**emory (**TM**). A group of $N_p$ learnable prompt tokens $\mathbf{p} = [p_1, ..., p_{N_p}] \in$

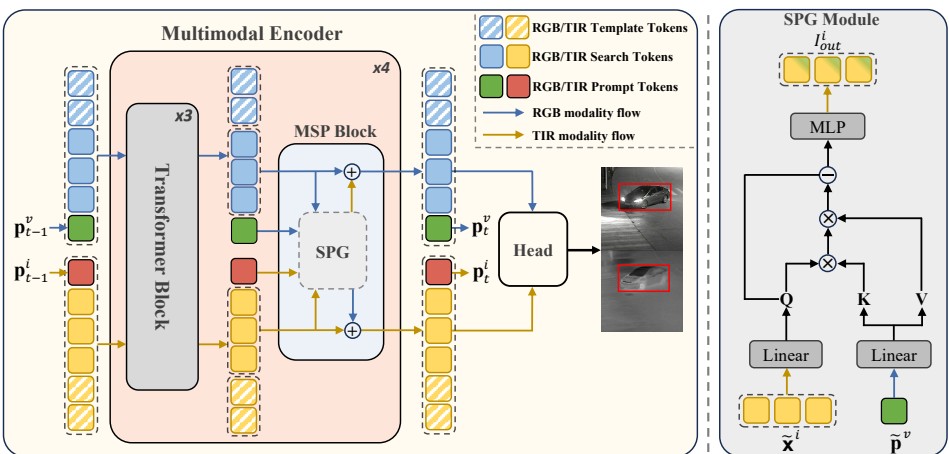

Figure 3: **Model structure of PromptTrack**. It employs a multimodal encoder for spatial-temporal relationship modeling by inserting the proposed MSP block between the transformer blocks. The extracted RGB and TIR search features from the encoder are fed into the head for target localization. The multimodal spatial prompt (SPG) module within the MSP block is illustrated on the right.

$\mathbb{R}^{N_p \times D}$ are incorporated into the input token sequence as $[\mathbf{z}_0; ...; \mathbf{z}_{t-1}; \mathbf{x}_t; \mathbf{p}_{t-1}]$, which interacts with all spatial-temporal image tokens ($[\mathbf{z}_0; ...; \mathbf{z}_{t-1}; \mathbf{x}_t]$) to learn temporal cues. During the relationship modeling of the encoder at timestep $t$, prompt tokens $\mathbf{p}_{t-1}$ from the last timestep ($t-1$) provide prior appearance and position information of the target for search tokens, and also aggregate appearance variations and motion trends at the current timestep ($t$). These continuously updated tokens, referred to as **streaming temporal prompt**, help the tracker to identify and locate the target across the timeline. It is worth noting that due to the different appearance variations of targets within individual RGB and TIR modalities, a distinct group of learnable tokens ($\mathbf{p}_0^v$ and $\mathbf{p}_0^i$) is designated for each modality at the initial timestep.

**Template sample.** Usually, the differences between adjacent template images are minimal, leading to significant information redundancy. Additionally, inputting all template images from TM into the encoder would result in unsustainable computational costs. Considering these factors, we design three different strategies to sample $k$ template images from TM: (1) Top-k score-based sampling. (2) Last-k sampling. (3) Uniform interval sampling.

Based on these sampling strategies, the input sequence of tokens is denoted as:

$$\mathbf{S} = [\mathbf{z}_{j_1}; ...; \mathbf{z}_{j_k}; \mathbf{x}_t; \mathbf{p}_{t-1}] \in \mathbb{R}^{(N_z * k + N_x + N_p) \times D}, \tag{2}$$

$$[\mathbf{z}_{j_1}; ...; \mathbf{z}_{j_m}; ...; \mathbf{z}_{j_k}] = \text{Sample}([\mathbf{z}_0; ...; \mathbf{z}_{t-1}]), j_m \in [0, t-1] \tag{3}$$

The details and comparative results of these sampling strategies can be seen in Section 4.4.

### 3.3 MULTIMODAL SPATIAL PROMPT LEARNING

Taking into account the spatial alignment characteristics of paired RGB and TIR search images at each timestep $t$ in the video stream, we propose a multimodal spatial prompt learning method to effectively leverage multimodal complementary information. Our key approach involves inserting our proposed **multimodal spatial prompt** (MSP) block between transformer blocks of the original encoder. This method constructs a multimodal encoder that effectively facilitates multimodal spatial interactions within the MSP block. The model structure of PromptTrack is illustrated in Figure 3. During the forward propagation of the encoder, intra-modal feature extraction within the transformer block and inter-modal interaction within the MSP block are performed iteratively. Ultimately, the extracted RGB and TIR search features (tokens) are fed into the head to predict the bounding box of the target, and the updated temporal prompt tokens are split for the next timestep.

**Multimodal encoder.** Given the initial RGB input token sequence $\mathbf{S}_0^v = [\mathbf{z}_{j_1}^v; ...; \mathbf{z}_{j_k}^v; \mathbf{x}_t^v; \mathbf{p}_t^v]$ and TIR input token sequence $\mathbf{S}_0^i = [\mathbf{z}_{j_1}^i; ...; \mathbf{z}_{j_k}^i; \mathbf{x}_t^i; \mathbf{p}_t^i]$, the forward process of the $l$-th transformer block

is formulated as follows:

$$\mathbf{S}_l^v = \mathcal{F}_l(\mathbf{S}_{l-1}^v), \mathbf{S}_l^i = \mathcal{F}_l(\mathbf{S}_{l-1}^i), l = 1, ..., 12 \tag{4}$$

where $\mathcal{F}_l$ represents the learnable mapping function of the $l$-th transformer block, which includes self-attention and a feed-forward network (FFN). We take the inserted MSP block after the $l$-th transformer block as an example to formulate the multimodal spatial prompt learning process. Firstly, we split $\mathbf{S}_l^v$ and $\mathbf{S}_l^i$ in the spatial dimension to obtain updated RGB search tokens $\tilde{\mathbf{x}}^v$, RGB prompt tokens $\tilde{\mathbf{p}}^v$, TIR search tokens $\tilde{\mathbf{x}}^i$, and TIR prompt tokens $\tilde{\mathbf{p}}^i$. Then we utilize our designed spatial prompt generation (**SPG**) module to learn multimodal spatial prompts. Next, the RGB and TIR search features are further enhanced by the generated TIR and RGB spatial prompts, respectively. Based on spatial prompt learning, the multimodal interaction can be formulated as follows:

$$\tilde{\mathbf{x}}_{out}^v = \tilde{\mathbf{x}}^v + SPG(\tilde{\mathbf{x}}^i, \tilde{\mathbf{p}}^v) \tag{5}$$

$$\tilde{\mathbf{x}}_{out}^i = \tilde{\mathbf{x}}^i + SPG(\tilde{\mathbf{x}}^v, \tilde{\mathbf{p}}^i) \tag{6}$$

Finally, the enhanced search token $\tilde{\mathbf{x}}_{out}$ are re-concatenated with template tokens and temporal tokens from the same modality for the next transformer block. We provide a comparison of model efficiency after applying the multimodal encoder versus the original encoder in Table 6.

**Spatial prompt generation.** The SPG module requires two input components: *search tokens from one modality and temporal prompt tokens from the other modality.* As illustrated in the right part of Figure 3, let's describe the generation process of TIR spatial prompt, using TIR search tokens ($\tilde{\mathbf{x}}^i$) and RGB prompt tokens ($\tilde{\mathbf{p}}^v$) as an example. The TIR search tokens act as query, while RGB prompt tokens serve as the key and value. The common information $I_{comm}^i$ from both modality is extracted by multi-head cross-attention mechanism as follows:

$$Q = Linear(\tilde{\mathbf{x}}^i), K = Linear(\tilde{\mathbf{p}}^v), V = Linear(\tilde{\mathbf{p}}^v) \tag{7}$$

$$I_{comm}^i = Softmax(\frac{Q \cdot K^T}{\sqrt{d}}) \cdot V \tag{8}$$

where $d$ represents the dimension of each head. Based on the argument that *common information (e.g., salient objects) can be effectively captured through joint template-search-prompt relationship modeling within the transformer block* (Cui et al., 2024), we further obtain TIR modality-specific information $I_{spec}^i$ by removing the common information. Subsequently, the TIR spatial prompt $I_{out}^i$ is output by an MLP layer. The process can be defined as:

$$I_{spec}^i = Q - I_{comm}^i \tag{9}$$

$$I_{out}^i = MLP(I_{spec}^i) \tag{10}$$

Similarly, RGB spatial prompt $I_{out}^v$ is generated through the above operations to provide complementary information for TIR search tokens. Compared to static spatial prompts in ViPT (Jiawen et al., 2023), our multimodal spatial prompts are conditioned on temporal prompts with minimal computational overhead and are thus more effective to focus on targets in complex tracking scenarios.

### 3.4 HEAD AND LOSS

We adopt the same structure of the Center head as described in OSTrack (Ye et al., 2022) to predict the bounding box, which consists of stacked Conv-BN-ReLU layers. The overall loss function is:

$$\mathcal{L} = \mathcal{L}_{cls} + \lambda_{GIoU}\mathcal{L}_{GIoU} + \lambda_{L1}\mathcal{L}_{L1} \tag{11}$$

where $\mathcal{L}_{cls}$ represents the weighted focal loss (Law & Deng, 2018) for classification, $\mathcal{L}_{GIoU}$ denotes the generalized IoU loss (Rezatofighi et al., 2019), and $\mathcal{L}_{L1}$ corresponds to the bounding box regression loss. Additionally, the weighted factors $\lambda_{GIoU}$ and $\lambda_{L1}$ are set to 2 and 5, respectively.

## 4 EXPERIMENTS

### 4.1 IMPLEMENTATION DETAILS

We adopt the ViT-Base (Kolesnikov et al., 2021) as the original encoder, which is pretrained on popular single object tracking datasets (SOT) (Ye et al., 2022). Our MSP block is inserted after the

3-rd, 6-th, 9-th, and 12-th blocks of the encoder for multimodal interaction. The template image size is $128 \times 128$ and the search region size is $256 \times 256$. The number of learnable temporal tokens $Np$ for each modality are set to 4, corresponding to the variables needed to represent a bounding box. We crop the new template image centered on the predicted box and push it into the historical template queue. The weights of the transformer blocks and head are initialized with the pretrained OSTrack-256 model (Ye et al., 2022). The inserted blocks and learnable temporal tokens are initialized with random weights. The total batch size is 16. The learning rate is set to $4 \times 10^{-5}$ for the transformer blocks and head, and $4 \times 10^{-6}$ for the remaining components. We use the AdamW optimizer (Loshchilov & Hutter, 2017) with weight decay of $1 \times 10^{-4}$. Horizontal flip and brightness jittering are used for data augmentation.

**Training.** The training process requires 20 epochs based on the LasHeR training set (Li et al., 2021), with each epoch comprising $40k$ samples. The process is divided into two stages: the first stage involves training for 10 epochs to learn the relationship modeling between image tokens and initial temporal tokens, with each sample containing $k$ template images and 1 search image. The second stage consists of another 10 epochs, where each sample contains $k$ template images and 2 search images, focusing on learning temporal information propagation conditioned on the updated temporal tokens. The sample interval is set to 400 within a single video sequence. All training stages are performed on two NVIDIA 3090 GPUs using Python 3.9, Pytorch 2.0.0, and CUDA 11.7. The same training configuration is employed across all experiments, including ablation studies. The inference speed (FPS) is evaluated on one NVIDIA 3090 GPU, 12th Gen Intel(R) Core(TM) i9-12900K CPU and 64GB of memory.

## 4.2 METRICS FOR RGB-T TRACKING

We adopt two widely used metrics, Precision rate (PR) and Success rate (SR), to evaluate tracking performances. PR is the percentage of video frames in which the Euclidean distance between the center coordinates of the predicted box and the ground truth (GT) within a certain threshold (typically 20 pixels). SR is the proportion of frames where the Intersection over Union (IoU) between the predicted bounding box and the ground truth exceeds a predefined overlap threshold.

## 4.3 COMPARISONS WITH STATE-OF-THE-ART TRACKERS

Table 1: State-of-the-art comparison on LasHeR, RGBT210 and RGBT234 datasets. The top two results are highlighted in **bold** and underline fonts. Results are reported in percentage (%).

| Tracker | Backbone | Pretrain | Temporal | LasHeR | | RGBT210 | | RGBT234 | | FPS |
|---|---|---|---|---|---|---|---|---|---|---|
| | | | | PR(↑) | SR(↑) | PR(↑) | SR(↑) | PR(↑) | SR(↑) | |
| DAFNet (Gao et al., 2019) | VGG-M | ImageNet | ✗ | 44.9 | 31.1 | - | - | 79.6 | 54.4 | 20.5 |
| MANet (Long Li et al., 2019) | VGG-M | ImageNet | ✗ | 45.5 | 32.6 | - | - | 77.7 | 53.9 | 2.1 |
| CAT (Li et al., 2020) | VGG-M | ImageNet | ✗ | 45.1 | 31.7 | 79.2 | 53.3 | 80.4 | 56.1 | 20 |
| CMPP (Wang et al., 2020) | VGG-M | ImageNet | ✗ | - | - | - | - | 82.3 | 57.5 | 1.3 |
| MANet++ (Lu et al., 2021) | VGG-M | ImageNet | ✗ | 46.7 | 31.7 | - | - | 80.0 | 55.4 | 25.4 |
| TFNet (Zhu et al., 2022) | VGG-M | ImageNet | ✗ | - | - | 77.7 | 52.9 | 80.6 | 56.0 | 17 |
| MFGNet (Wang et al., 2022) | VGG-M | ImageNet | ✗ | - | - | 74.9 | 49.4 | 78.3 | 53.5 | 23 |
| APFNet (Xiao et al., 2022) | VGG-M | ImageNet | ✗ | 50.0 | 36.2 | - | - | 82.7 | 57.9 | 1.9 |
| OSTrack (Ye et al., 2022) | ViT | SOT | ✗ | 51.5 | 41.2 | - | - | 72.9 | 54.9 | 45.5 |
| QAT (Liu et al., 2023) | ResNet-50 | SOT | ✗ | 64.2 | 50.1 | 86.8 | 61.9 | 88.4 | 64.3 | - |
| ViPT (Jiawen et al., 2023) | ViT | SOT | ✗ | 65.1 | 52.5 | - | - | 83.5 | 61.7 | - |
| TBSI (Hui et al., 2023) | ViT | SOT | ✗ | 69.2 | 55.6 | 85.3 | 62.5 | 87.1 | 63.7 | 36.2 |
| BAT (Cao et al., 2024) | ViT | SOT | ✗ | 70.2 | 56.3 | - | - | 86.8 | 64.1 | - |
| TAAT (Tang et al., 2022) | ResNet-50 | SOT | ✓ | 55.9 | 34.4 | 78.6 | 55.5 | 78.5 | 44.1 | - |
| DMSTM (Zhang et al., 2023) | VGG-M | ImageNet | ✓ | 55.7 | 40.0 | - | - | 78.6 | 56.2 | 27.6 |
| STMT (Sun et al., 2024) | ViT | SOT | ✓ | 67.4 | 53.7 | 83.0 | 59.5 | 86.5 | 63.8 | 39.1 |
| TATrack (Wang et al., 2024) | ViT | SOT | ✓ | 70.2 | 56.3 | 85.3 | 61.8 | 87.2 | 64.4 | - |
| **PromptTrack** | ViT | SOT | ✓ | **76.2** | **60.7** | **90.6** | **66.1** | **91.7** | **67.2** | 35 |

**LasHeR** (Li et al., 2021) is the most challenging dataset in the RGB-T tracking domain, including complex scenarios such as similar distractors and long-term tracking. It consists of 1224 pairs of visible light and thermal infrared video sequences, totaling over 730k frames. As shown in Table 1, our PromptTrack significantly outperforms the state-of-the-art non-temporal tracker BAT (Cao et al., 2024) by 6% in PR and 4.4% in SR. The substantial improvement in performance indicates that

integrating temporal information during multimodal interaction enhances perception of targets in complex tracking environments. In comparison with temporal-based methods relying solely on dynamic templates, our approach surpasses STMT (Sun et al., 2024) and TATrack (Wang et al., 2024) by 8.8% and 6.0% in PR, and 7.0% and 4.4% in SR, respectively. This highlights that our method fully leverages rich and dense spatial-temporal cues to enhance target localization capabilities.

**RGBT210** (Li et al., 2017) is a popular tracking benchmark with 150 short-term video sequences. Compared to the ResNet-based (He et al., 2016) tracker QAT (Liu et al., 2023), our method surpasses it by 3.8% in PR and 4.2% in SR respectively, without any fine-tuning.

**RGBT234** (Li et al., 2019) is an extension of RGBT210 with 24 additional sequences, which provides 12 attributes for a comprehensive evaluation of trackers. From the results, it can be observed that PromptTrack achieves the best performance, with PR and SR scores of 91.7% and 67.2% respectively. This represents an improvement of 8.2% and 3.3% in PR, 5.5% and 2.9% in SR over ViPT (Jiawen et al., 2023) and BAT (Cao et al., 2024), which are also based on our prompt learning paradigm. This indicates that utilizing streaming spatial-temporal prompts across historical frames enables the model to possess more robust capabilities compared to relying solely on spatial prompts.

## 4.4 ABLATION STUDY

To verify the effectiveness of our proposed method, we investigate different designs of PromptTrack and perform comprehensive ablation studies on LasHeR and RGBT234 datasets.

Table 2: Ablation study on different prompt settings.

| # | Setting | | LasHeR | | RGBT234 | |
|---|---|---|---|---|---|---|
| | temporal prompt | spatial prompt | PR | SR | PR | SR |
| ① | | | 71.9 | 57.6 | 89.5 | 64.6 |
| ② | ✓ | | 74.8 | 59.7 | 91.0 | 66.4 |
| ③ | | ✓ | 73.8 | 58.9 | 90.8 | 65.7 |
| ④ | ✓ | ✓ | **76.2** | **60.7** | **91.7** | **67.2** |

Table 3: Comparison of different template sampling methods on LasHeR.

| Strategy | PR | SR |
|---|---|---|
| Top-k score sampling | 71.9 | 57.6 |
| Last-k sampling | 74.2 | 59.3 |
| **Uniform interval sampling** | **76.2** | **60.7** |

**Impact of different prompt settings.** In Table 2, ① indicates that we only employ a modality-shared transformer encoder to respectively extract RGB and TIR modal features without temporal and spatial prompts for multimodal interaction. The extracted RGB and TIR search features are directly concatenated along the channel dimension and then fed into the head. The results on LasHeR still surpass those temporally-sparse trackers. The findings indicate that historical templates can effectively enhance the model's ability to discriminate targets in challenge tracking environments. Notably, while incorporating streaming temporal prompts, the performances of the model are significantly enhanced (①$vs.$②). Experimental results suggest that leveraging dense temporal associations to provide prior information about the target is crucial for the video-level RGB-T tracking task.

Due to the absence of temporal prompts in ③, we map RGB search tokens as the key and value in Equation (7) to generate TIR spatial prompt. The performance improvement compared to ① demonstrates that our designed MSP block can effectively leverage complementary information for multimodal interaction. However, the significantly larger number of search tokens compared to temporal tokens ($256 vs. 4$) results in higher computational cost, making the use of temporal tokens more efficient. PromptTrack (④), involving streaming spatial-temporal prompts, exhibits the most advanced tracking performance. The multimodal spatial prompt generation conditioned on temporal information allows the model to focus on target-related information, eliminating the interference of background noise from search regions.

**Impact of template sampling strategies.** To verify the impact of different template sampling methods on performance *when inference*, we compared three different sampling techniques. The default number $k$ of sampled templates is set to 4. As shown in Table 3, *Top-k score sampling* means selecting the top k template images with the highest confidence scores from the template memory (TM). This method causes the sampled templates to be concentrated in the initial segments of the video and in simpler scenes, leading to much lower performance. *Last-k sampling* denotes selecting the last consecutive k templates stored in the TM, which is lower than *uniform interval sampling* (i.e., sampling at equal time intervals). We choose the uniform interval sampling as the default setting, which provides consistent historical templates in both simple and challenging scenes.

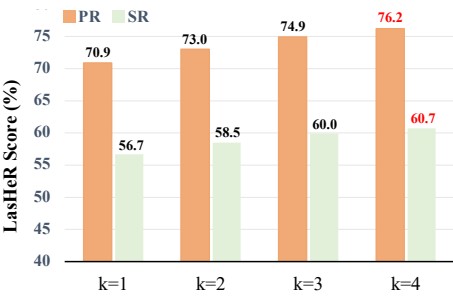

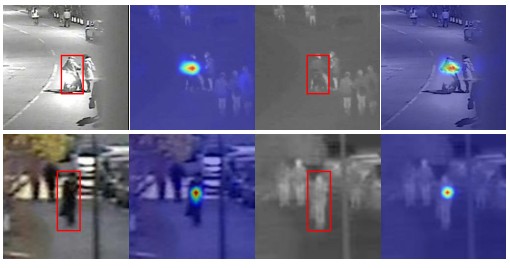

(a) RGB search    (b) RGB attention map    (c) TIR search    (d) TIR attention map

Figure 4: Comparison of different numbers of historical templates.

Figure 5: Attention visualization from temporal tokens to search tokens. The red boxes denote the GT.

**Impact of historical templates.** The results are shown in Figure 4. As the number of historical templates increases, tracking performance gradually improves. In this experiment, the same number of templates is used for both training and inference. However, due to computational resource limitations, we set a maximum of 4 templates. Future work could explore the use of more templates, which we speculate would further enhance performance.

**Visualization.** To gain deep insights into temporal tokens, we generate cross-attention maps about the temporal tokens to the search tokens, where the temporal tokens are served as query, as illustrate in Figure 5. The visualization results indicate that temporal tokens precisely focus on the target, even in the presence of similar distractors. This can be attributed to temporal tokens also learning information about motion trajectory of the target from historical templates. More attention visualization results can be seen in Figure 7 in the appendix.

## 4.5 EXTENDED EXPERIMENT FOR RGB-D TRACKING.

To demonstrate the versatility of PromptTrack across different domains such as RGB-Depth (RGB-D) Tracking, we conduct independent training and testing on the DepthTrack dataset without any model structure adjustments. As shown in Table 4, the performance of PromptTrack (RGB-D) surpass other top-performing trackers by a significant margin, demonstrating the excellent generality of our framework for multimodal tracking in other domains. More extended experiments of RGB-D and RGB-E can be seen in Table 7 and Table 8 in the appendix.

Table 4: Comparison of state-of-the-art RGB-D trackers on the DepthTrack test set.

|  | DeT (Yan et al., 2021) | OSTrack (Ye et al., 2022) | SPT (Zhu et al., 2023) | ProTrack (Yang et al., 2022) | ViPT (Jiawen et al., 2023) | OneTracker (Hong et al., 2024) | **PromptTrack** (RGB-D) |
|---|---|---|---|---|---|---|---|
| F-score | 53.2 | 52.9 | 53.8 | 57.8 | 59.4 | 60.9 | **64.4** |
| Re | 50.6 | 52.2 | 54.9 | 57.3 | 59.6 | 60.4 | **64.3** |
| Pr | 53.6 | 53.6 | 52.7 | 58.3 | 59.2 | 60.7 | **64.5** |

## 5 CONCLUSION

In this paper, we have presented a novel video-level RGB-T tracking paradigm via prompt learning, which learns rich temporal cues and complementary spatial information across consecutive frames. Our PromptTrack significantly improves performance in complex tracking scenarios by incorporating streaming temporal prompts about appearance changes and motion trajectories of targets from historical templates. The multimodal spatial information is efficiently utilized conditioned on the temporal information, eliminating the need for complex spatial fusion module designs. The prompt-based framework can be also extended to other multimodal tracking domains. We hope PromptTrack can facilitate the exploration of spatial-temporal information for multimodal tracking in the future.

**Limitations.** One limitation of our method is the requirement for substantial storage resources to retain all historical templates, as well as significant computational resources to perform interactions between all input images. It would be interesting to explore the use of extracted target tokens for storage and computation.

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

APPENDIX

## A   INFERENCE DETAILS

The template memory (TM) size is not fixed during inference and can theoretically grow without bounds. At timestep $t$, it stores all templates from timestep 0 to $t-1$. After tracking the target at timestep $t$, a new template is cropped based on the predicted bounding box and subsequently stored in the TM. For uniform interval sampling at timestep $t$, all templates from TM are divided into $k$ segments of equal length $l_{\text{seg}}$, and we sample the middle frame from each segment. According to Equation (3), the sampled indices are denoted as $j_m$, with the calculation formulas as follows:

$$l_{\text{seg}} = \left\lfloor \frac{t}{k} \right\rfloor \tag{12}$$

$$j_m = (m-1) \cdot l_{\text{seg}} + \left\lfloor \frac{l_{\text{seg}}}{2} \right\rfloor, \quad m = 2, 3, \ldots, k, \quad \text{where } t \geq k \tag{13}$$

To ensure that the supervision information from the initial frame is preserved, the sampled $k$ templates will always include the first frame template, i.e., $j_1 = 0$. When $t < k$, all templates in the TM are used. In Figure 8 (c), we present a visualization of the sampled templates with frame id indicated, using the uniform interval sampling strategy. The overall effect shows that this strategy samples templates over a longer time span, thereby optimizing performance on long-term sequences.

## B   COMPARISON OF MODALITY-SPECIFIC AND COMMON INFORMATION

Compared to the traditional cross-attention operation, we utilize Equation (9) to obtain modality-specific information $I_{spec}^i$ when generating TIR spatial prompts, rather than information common $I_{comm}^i$ between two modalities. To demonstrate the advantage of this approach, we conduct comparative experiments, as shown in Table 5. Our method achieves a performance improvement, with increases of 0.9% in PR and 0.4% in SR on LasHeR, which indicates that modality-specific information is more important for spatially aligned RGB and TIR images. We hope this idea will also provide researchers with new insights into studies of multimodal complementary information.

Table 5: Modality-specific and common information study.

| Setting | LasHeR | | RGBT234 | |
|---|---|---|---|---|
| | PR | SR | PR | SR |
| w/ common information | 75.3 | 60.3 | 91.1 | 66.8 |
| **w/ specific information** | **76.2** | **60.7** | **91.7** | **67.2** |

## C   COMPARISON OF MODEL PERFORMANCE AND EFFICIENCY

We conduct a comparative analysis of model performance and efficiency on the LasHeR dataset. The results are shown in Table 6. *OSTrack (RGB-T)* denotes our baseline model, derived by extending the encoder of the *OSTrack (RGB)* tracking model into a modality-shared encoder to separately process inputs from RGB and TIR modalities. The increase in the number of parameters (+56.7M) comes from the *inserted four MSP blocks and temporal tokens*. The increase in Multiply–Accumulate Operations (MACs) arises from the *spatial-temporal interactions with $k$ historical templates and multimodal spatial prompt generation modules*. Due to the need to store all historical template images of $128 \times 128$ size, memory consumption varies with different sequence lengths. For instance, the longest sequence *blkboyshead* from the LasHeR test set, which consists of 12862 frames, requires 12GB of memory and 1587MB of GPU memory. Despite the higher computational overhead, PromptTrack significantly outperforms the baseline model in terms of PR and SR on LasHeR while still runing at a real-time speed of 35 FPS.

Table 6: Comparison of model performance and efficiency on LasHeR.

| Tracker | Performance | | Efficiency | | |
|---------|:---:|:---:|:---:|:---:|:---:|
| | PR | SR | Params | MACs | FPS |
| OSTrack (RGB) | - | - | 92.1M | 29.0G | 130 |
| OSTrack (RGB-T) | 51.5 | 41.2 | 102.7M | 59.8G | 69 |
| **PromptTrack** (RGB-T) | **76.2** | **60.7** | 159.4M | 104.8G | 35 |

# D ATTRIBUTE-BASED PERFORMANCE ANALYSIS

To evaluate the performance of PromptTrack in various scenarios, we compare it with the current state-of-the-art methods based on 12 challenge attributes from the RGBT234 dataset, including No Occlusion, Partial Occlusion, Heavy Occlusion, Low Illumination, Low Resolution, Thermal Cross, Object Deformation, Fast Motion, Scale Variation, Motion Blur, Camera Motion, and Background Clutter. As shown in Figure 6, PromptTrack achieves the best performance across all challenge attributes, especially on **Scale Variation**, **Fast Motion**, and **Motion Blur** attributes. This indicates the strong robustness of PromptTrack against many complex challenges, as it can densely model spatial-temporal associations of targets across consecutive frames.

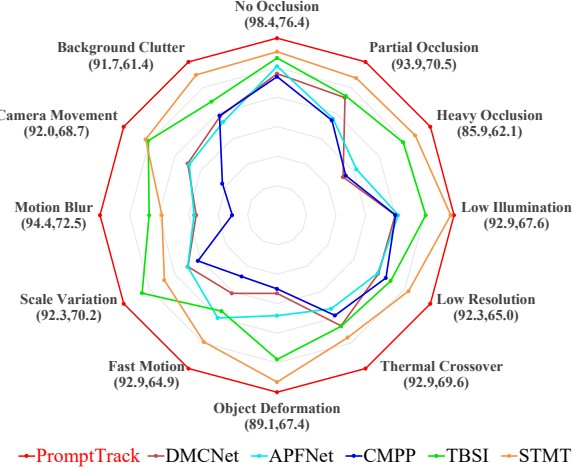

Figure 6: Attribute-based Performance of PR/SR scores on RGBT234.

# E EXTENDED STUDIES ON RGB-D AND RGB-E TRACKING DOMAINS

To further demonstrate the generality of PromptTrack in different domains such as RGB-D and RGB-Event (RGB-E), we conduct additional experimental exploration on VOT-RGBD2022 (Kristan et al., 2022) and VisEvent (Wang et al., 2023) without any model structure modification. For RGB-D tracking, we train the model on the DepthTrack training set and test it on the DepthTrack test set and VOT-RGBD2022 dataset. For RGB-E tracking, we train the model on the VisEvent training set and test it on the VisEvent test set.

**DepthTrack** (Yan et al., 2021) is a large-scale long-term RGB-D tracking dataset consisting of 150 training videos and 50 testing videos, with an average of 1473 frames per video. The results in the RGB-D tracking domain are shown in Table 4, PromptTrack (RGB-D) outperforms recent state-of-the-art methods ViPT and OneTracker by **5%** and **3.5%** in terms of F-score.

**VOT-RGBD2022** (Kristan et al., 2022) is the latest RGB-D benchmark dataset, comprising 127 sequences designed for leveraging depth information in RGB-D tracking. The dataset adopts an anchor-based short-term evaluation protocol, which requires trackers to initiate from various initialization points for multiple starts. The expected average overlap (EAO) is the overall performance

metric. As shown in Table 7, our PromptTrack (RGB-D) outperforms previous RGB-D methods, achieving an EAO of **77.6%**, which is a **4.9%** improvement in EAO compared to the OneTracker.

**VisEvent** (Wang et al., 2023) is currently the first large-scale benchmark dataset for RGB-E tracking collected from the real world. As shown in Table 8, PromptTrack (RGB-E) achieves 76.5% in PR and 62.6% in SR, surpassing other state-of-the-art RGB-E trackers.

The superior performances across different multimodal tracking domains demonstrate the effectiveness and generality of PromptTrack, indicating its capability to learn effective spatial-temporal prompts for guiding the localization of multimodal targets.

Table 7: Comparison of state-of-the-art RGB-D trackers on VOT-RGBD2022.

| | DeT (Yan et al., 2021) | OSTrack (Ye et al., 2022) | SPT (Zhu et al., 2023) | ProTrack (Yang et al., 2022) | ViPT (Jiawen et al., 2023) | OneTracker (Hong et al., 2024) | **PromptTrack** (RGB-D) |
|---|---|---|---|---|---|---|---|
| EAO | 65.7 | 67.6 | 65.1 | 65.1 | 72.1 | 72.7 | **77.6** |
| Accuracy | 80.3 | 80.3 | 79.8 | 80.1 | 81.5 | **81.9** | 81.7 |
| Robustness | 83.3 | 83.3 | 85.1 | 80.2 | 87.1 | 87.2 | **94.2** |

Table 8: Comparison of state-of-the-art RGB-E trackers on the VisEvent test set.

| | ProTrack (Yang et al., 2022) | TransT (Chen et al., 2021) | LTMU (Dai et al., 2020) | OSTrack (Ye et al., 2022) | ViPT (Jiawen et al., 2023) | OneTracker (Hong et al., 2024) | **PromptTrack** (RGB-E) |
|---|---|---|---|---|---|---|---|
| PR | 63.2 | 65.0 | 65.5 | 69.5 | 75.8 | **76.7** | 76.5 |
| SR | 47.1 | 47.4 | 45.9 | 53.4 | 59.2 | 60.8 | **62.6** |

# F  MORE ATTENTION VISUALIZATION

We provide additional visualization results of attention maps from temporal tokens to search tokens for two representative video sequences selected from the LasHeR dataset, as shown in Figure 7. In the *moto* sequence, it can be seen that even in cases of target occlusion and TIR target blurring, the temporal tokens are able to effectively focus on the target, benefiting from the learned motion trajectory of the target across consecutive frames. In the *11leftboy* sequence, the temporal tokens also provide consistent attention despite appearance changes of the target. This visualization results fully demonstrate our streaming temporal prompts can learn information about the target's appearance changes and trajectory.

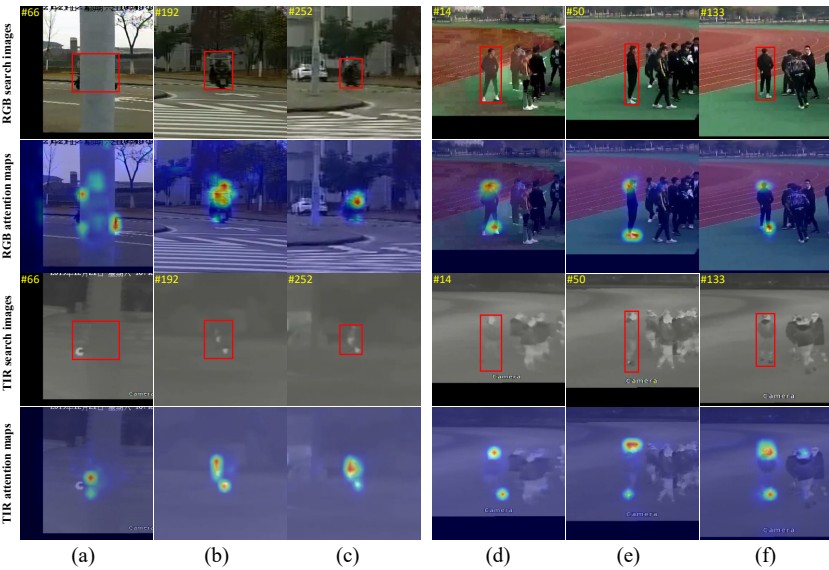

Figure 7: Attention visualization across different timesteps. The red boxes denote the GT. (a)-(c): RGB and TIR search images and corresponding attention maps of the *moto* sequence. (d)-(f): RGB and TIR search images and corresponding attention maps of the *11leftboy* sequence.

# G    QUALITATIVE EVALUATION

To intuitively demonstrate the effectiveness of our method, we compare PromptTrack with some state-of-the-art methods and visualize tracking results in various challenging scenarios. We select some representative sequences from RGBT234, involving occlusion, low illumination, similar distractors, and long-term tracking. As shown in Figure 8, our PromptTrack exhibits excellcent tracking precision and robustness. For instance, in scenarios (a) and (d), *where the target is occluded or undergoes deformation due to long-term tracking*, PromptTrack can still accurately locate the target by leveraging the temporal information provided by temporal prompts. In the scenario (b), *under low illumination conditions*, PromptTrack utilizes the complementary spatial information through generating multimodal spatial prompts to stably track multimodal targets. These results indicate that our proposed method effectively addresses many challenge challenges, enhancing the tracker's discriminative power.

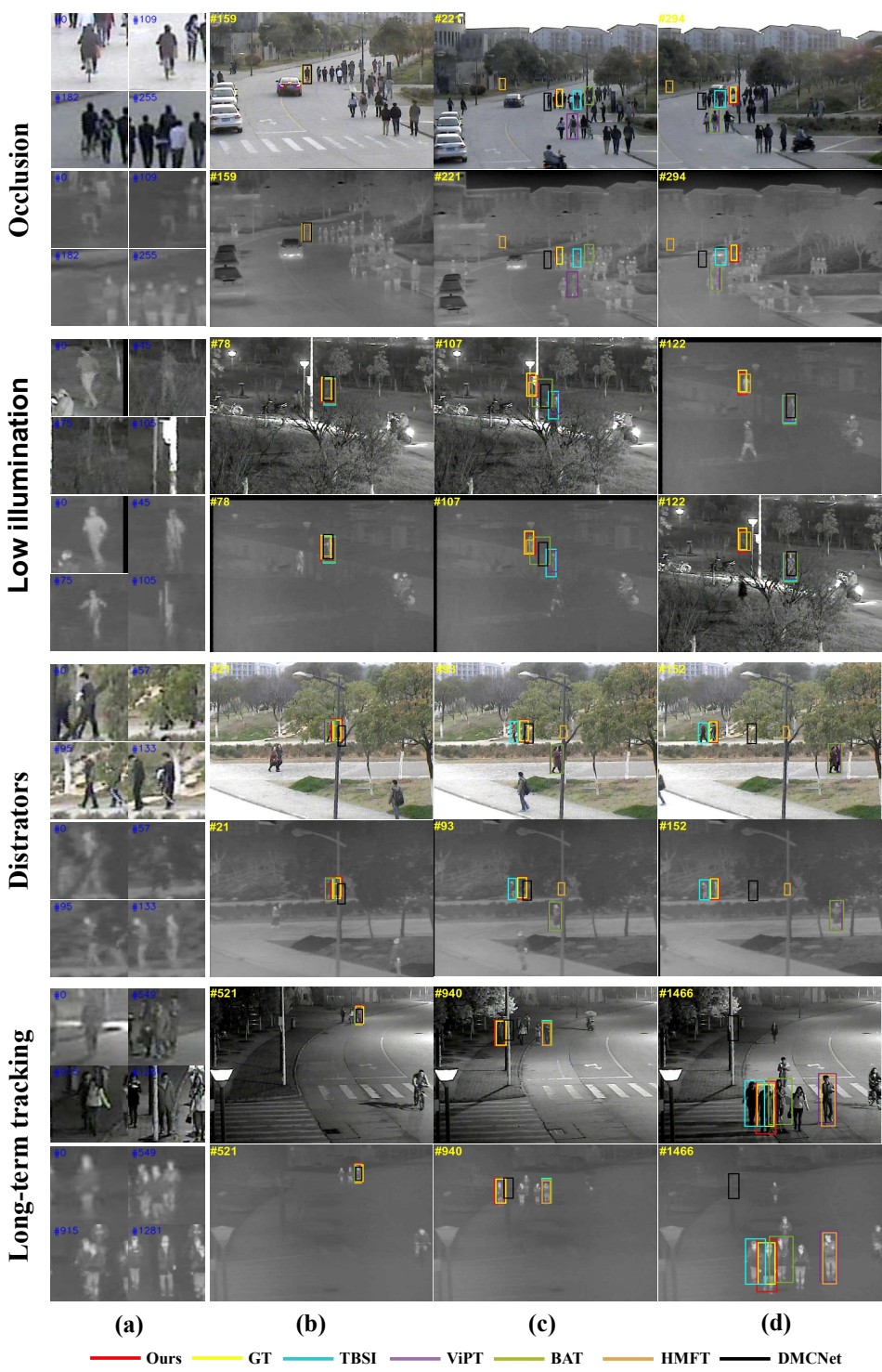

Figure 8: Qualitative comparison results of our tracker with other SOTA trackers on four representative sequences from the RGBT234 dataset. (a): Sampled historical template images with frame id from TM. (b)-(d): Tracking results at different timesteps.

