# OpenReview forum: "Streaming Spatial-Temporal Prompt Learning for RGB-T Tracking"
_ICLR.cc/2025/Conference — Submitted to ICLR 2025_

### Official Review · Reviewer_QfPF · 2024-10-22

**Soundness:** 2
**Presentation:** 3
**Contribution:** 2
**Rating:** 3
**Confidence:** 5

**Summary:**

This paper introduces PromptTrack, a novel video-level RGB-T tracking paradigm leveraging prompt learning to model spatiotemporal relationships in multimodal contexts. The authors argue that many existing methods inadequately utilize temporal information, often focusing on spatial aspects or introducing sparse temporal cues. In contrast, PromptTrack employs spatiotemporal prompts, allowing it to track target appearance changes and motion trajectories more effectively across video frames.

By learning temporal prompts for each modality and then incorporating multimodal spatial prompts conditioned on these temporal prompts, the proposed method enhances the complementary use of multimodal data.

Experimental results show that PromptTrack achieves state-of-the-art performance on several benchmark datasets, including a Precision score of 76.2% and a Success score of 60.7% on the LasHeR dataset, while maintaining a real-time speed of 35 FPS.

**Strengths:**

1.PromptTrack captures target appearance changes and motion trajectories by incorporating streaming spatiotemporal prompts, resulting in more accurate and robust tracking across video frames.

2.It achieves state-of-the-art tracking results while maintaining real-time speeds (35 FPS), demonstrating both high accuracy and practicality for real-world applications.

**Weaknesses:**

1.I believe the contribution of this paper is quite weak. It essentially adds an extra modality to ODTrack [1]. Regarding the claims made in its motivation, I even think it is almost indistinguishable from ODTrack. Moreover, like ODTrack, it also employs a template sampling strategy.

2.The paper employs a memory mechanism. For a fairer comparison, I believe the authors should clearly indicate which papers use a memory mechanism and which do not. From what I know, the use of a memory mechanism can greatly enhance tracking performance, and ODTrack also benefits from its memory mechanism. However, adding a memory mechanism typically reduces FPS, so it is necessary to clearly list this information.

[1] Zheng Y, Zhong B, Liang Q, et al. Odtrack: Online dense temporal token learning for visual tracking[C]//Proceedings of the AAAI Conference on Artificial Intelligence. 2024, 38(7): 7588-7596.

**Questions:**

1.The entire paper is built upon ODTrack, meaning the motivation proposed by the authors has already been addressed by others. What, then, is the authors' contribution?

---

### Official Review · Reviewer_2eSc · 2024-10-31

**Soundness:** 3
**Presentation:** 3
**Contribution:** 3
**Rating:** 6
**Confidence:** 4

**Summary:**

The paper proposes a spatial-temporal feature based RGBT tracking framework using prompt learning. The authors argue that current RGBT tracking methods do not make good use of temporal cues in videos, so they propose a streaming framework to continuously learn and utilize these features. The proposed prompt generation module can implicitly obtain the motion information in consecutive frames and propagate it to the next frame.

**Strengths:**

[1] The proposed prompt learning based temporal information learning model is an effective exploration of motion information integration for tracking.
[2] The framework is simple and effective, and experimental results on several benchmarks show that it achieves state-of-the-art performacne.

**Weaknesses:**

[1] There is need to explain carefully why the propsoed prompt is suitable to learn motion features.
[2] It is better to give a more detail comparison and analysis of how to intergrate the propsoed module in OSTrack.

**Questions:**

[1] How to get p_{0}  in Fig.2?
[2] Why we need to use a subtraction to get TIR spatial prompt in Eq.9 ? Is this operation improtant?

**Details Of Ethics Concerns:**

none.

---

### Official Review · Reviewer_Vy4K · 2024-11-04

**Soundness:** 3
**Presentation:** 2
**Contribution:** 3
**Rating:** 6
**Confidence:** 5

**Summary:**

This paper proposes a video-level RGB-T tracking paradigm called PromptTrack through prompt learning. It leverages temporal information from consecutive video frames to obtain temporal prompts, and learns multimodal spatial prompts conditioned on these temporal prompts, effectively utilizing complementary information from multiple modalities. A multimodal spatial prompt (SPG) module is inserted into the one-stream backbone network to enhance inter-modal interaction, while specific prompts are generated for each modality to capture modality-specific features. The current prompt tokens are stored in template memory for future use to enhance temporal information. The method has been evaluated on several benchmarks and achieves state-of-the-art performance.

**Strengths:**

1. Existing template search matching methods often utilize only spatial information for matching or merely introduce dynamic templates, neglecting the rich temporal cues present in consecutive video frames. The proposed PromptTrack establishes effective spatiotemporal associations by merging spatiotemporal flow prompts in the multimodal interaction process, enhancing robustness.

2. The motivation of the paper is clear, the experiments are comprehensive, and the results are convincing.

**Weaknesses:**

1. Various operations in Figure 3 should have annotations; otherwise, they may lead to ambiguity (e.g., it is not immediately clear whether the plus sign indicates concatenation or addition).

2. The authors mention one limitation of the method is the need for substantial storage resources to retain all historical templates, but they do not provide detailed information about the computational and memory costs during training and inference. More details about the computational and memory requirements during training and inference compared to baseline methods should be added.

3. The implementation details of the template memory are not sufficiently clear; it states that the maximum number of templates is four, but does not describe the size of the template memory or how the templates are updated. The paper should provide more detailed information about the template memory.

4. The concept of learnable prompt tokens is similar to the temporal token concept in previous works, such as ODTrack: Online Dense Temporal Token Learning for Visual Tracking, and what are the main differences between them?

5. In the experiments on RGBE and RGBD, the comparison trackers are limited to a few recent ones. To demonstrate the advance of the proposed method, I suggest more latest trackers should be added in comparison.

**Questions:**

Some details should be improved for clarity, and experimental comparison should be enhanced.

---

### Official Review · Reviewer_W7t5 · 2024-11-04

**Soundness:** 3
**Presentation:** 4
**Contribution:** 3
**Rating:** 6
**Confidence:** 5

**Summary:**

This paper proposes PromptTrack, a video-level RGB-T tracking paradigm via prompt learning (streaming temporal prompt and multimodal spatial prompt), achieving SOTA tracking performance and exhibits great scalability by extending to RGB-D and RGB-E tasks.

**Strengths:**

1. The reasonable usage of letters and formulas makes the entire text very readable and smooth.
2. Benefiting from the effective spatial-temporal associations during multimodal interaction, PromptTrack learns target changes and motion trajectory from dense historical frame and behaves better in complex environments.
3. In RGB-T/RGB-D/RGB-E benchmarks, the proposed method gains the advanced performance, verifying its effectiveness and versatility.

**Weaknesses:**

1. The ablation study about number of search images during training is missing. Does it increase would improve the temporal modeling and association capacity of the PromptTrack?
2. It's better to have an ablation study about number of MSP blocks and trade-off between performance and MACs/Parameters.
3. the "k" used in "comprising 40k samples" has a repeat appearance in the latter description like "k " template images. The authors should notice and improve this.

**Questions:**

1. The performace of OSTrack (RGB-T) in Table 6 of appendix.C is quite low, do you test it with fine-tuning or without any re-training?
2. The performance on three RGB-T datasets is based on single-set model parameters or divided sets? And will codes and pre-trained models soon release?

---

### Meta-Review · Area_Chair_b89W · 2024-12-21

**Metareview:**

The authors have not addressed the issues highlighted by the reviewers, and as a result, the manuscript will be rejected.

**Additional Comments On Reviewer Discussion:**

The authors have not addressed the issues highlighted by the reviewers, and as a result, the manuscript will be rejected.

---

### Decision · Program_Chairs · 2025-01-22

Reject